# Importance of the Use of Oxidative Stress Biomarkers and Inflammatory Profile in Aqueous and Vitreous Humor in Diabetic Retinopathy

**DOI:** 10.3390/antiox9090891

**Published:** 2020-09-20

**Authors:** Ana Karen López-Contreras, María Guadalupe Martínez-Ruiz, Cecilia Olvera-Montaño, Ricardo Raúl Robles-Rivera, Diana Esperanza Arévalo-Simental, José Alberto Castellanos-González, Abel Hernández-Chávez, Selene Guadalupe Huerta-Olvera, Ernesto German Cardona-Muñoz, Adolfo Daniel Rodríguez-Carrizalez

**Affiliations:** 1Department of Physiology, Health Sciences University Center, Institute of Clinical and Experimental Therapeutics, University of Guadalajara, Guadalajara, Jalisco 44340, Mexico; Karen.LopezC@alumno.udg.mx (A.K.L.-C.); maria.mruiz@alumnos.udg.mx (M.G.M.-R.); cecilia.Olvera@alumno.udg.mx (C.O.-M.); raul.roblesr@alumnos.udg.mx (R.R.R.-R.); darevalo@hcg.gob.mx (D.E.A.-S.); jose.castellanos2223@alumnos.udg.mx (J.A.C.-G.); abel.hchavez@academicos.udg.mx (A.H.-C.); cardona@cucs.udg.mx (E.G.C.-M.); 2Department of Ophthalmology, Hospital Civil de Guadalajara “Fray Antonio Alcalde”, Guadalajara, Jalisco 44280, Mexico; 3Department of Ophthalmology, Specialties Hospital of the National Occidental Medical Center, Mexican Institute of Social Security, Guadalajara, Jalisco 44329, Mexico; 4Medical and Life Sciences Department, La Ciénega University Center, University of Guadalajara, Ocotlán, Jalisco 47810, Mexico; selene.huerta@academicos.udg.mx

**Keywords:** diabetic retinopathy, oxidative stress, antioxidants, biomarkers of diabetic retinopathy, metabolic memory, tear film, aqueous humor, vitreous humor

## Abstract

Diabetic retinopathy is one of the leading causes of visual impairment and morbidity worldwide, being the number one cause of blindness in people between 27 and 75 years old. It is estimated that ~191 million people will be diagnosed with this microvascular complication by 2030. Its pathogenesis is due to alterations in the retinal microvasculature as a result of a high concentration of glucose in the blood for a long time which generates numerous molecular changes like oxidative stress. Therefore, this narrative review aims to approach various biomarkers associated with the development of diabetic retinopathy. Focusing on the molecules showing promise as detection tools, among them we consider markers of oxidative stress (TAC, LPO, MDA, 4-HNE, SOD, GPx, and catalase), inflammation (IL-6, IL-1ß, IL-8, IL-10, IL-17A, TNF-α, and MMPs), apoptosis (NF-kB, *cyt-c*, and caspases), and recently those that have to do with epigenetic modifications, their measurement in different biological matrices obtained from the eye, including importance, obtaining process, handling, and storage of these matrices in order to have the ability to detect the disease in its early stages.

## 1. Introduction

The retina is a transparent tissue of the eye which has an intricate arrangement of neurons and also requires a highly complex circulation to meet metabolic requirements and the proper functioning of neurotransmission, phototransduction, and complex interaction of metabolites, vasoactive agents, and growth factors [1,2]. The central retinal artery passes through the optic nerve to ensure blood flow, and gas and nutrient exchange, while the central retinal vein is involved in the elimination of waste products [3]. The retinal vasculature is of great importance and within its physiological functions; the most important is to maintain the internal blood–retinal barrier (iBRB), which prevents nonspecific penetration of macromolecules into the retinal neuropile. The outer blood–retinal barrier (oBRB), formed between the tight junctions of retinal pigment cells, maintains ionic concentrations in the avascular region of the retina and the interstitial space for neurotransmission [3]. Retinal vascular dysfunction occurs shortly after the onset of diabetes and is characterized by impaired microvasculature and transport across the blood–retinal barrier playing an important role in the onset and progression of vascular lesions in diabetic retinopathy (DR) [2,4]. There is currently a wide range of treatments available for diabetes mellitus (DM) which has dramatically increased the lifespan of diabetic patients, but in turn gives time for clinically significant microvascular complications to develop [5]. Currently, there is a wide variety of effective treatments for DR, diagnosing the disease in its early stages helps prevent progression to blindness [5,6].

In this narrative review, we aim to approach various biomarkers associated with the development of diabetic retinopathy. In particular, our objective is to focus on the importance of molecules that are promising as detection tools and their measurement in different biological matrices obtained from the eye, in order to achieve an early disease detection or, ideally, even before the actual start of the DR. Articles in English were included that showed relevance both in preclinical and clinical stages of the DR. We take into account the articles that contribute to the discussion of the use of biomarkers of different nature to identify and estimate the stage of the disease in which patients with DR are, in addition to the use of different biological matrices obtained directly from the eyeball such as the tear, aqueous humor, and vitreous humor.

## 2. Diabetic Retinopathy

Among the 468 million people with diabetes mellitus worldwide, approximately 90 million suffer from some form of DR [7]. It is the number one cause of blindness in people between the ages of 27 and 75. The prevalence of DR is approximately 25% and 90% at 5 and 20 years, respectively, from its diagnosis. Furthermore, it is estimated that ~191 million people will be diagnosed with this microvascular complication by 2030 [8,9], and the number of DR patients whose vision is threatened will increase from 37.3 to 56.3 million. This disturbing prospect makes the DR a significant global public health and economic problem [10].

Chronic hyperglycemia is the main risk factor affecting DR, as part of its pathophysiology it has been shown to induce vascular endothelial dysfunction in the retina [11]. When this state persist, activation of other pathways occurs in addition to glycolysis (such as polyol, hexosamine, and advanced glycation), which are known to induce apoptosis and pericyte degeneration, eventually damaging the retina [12].

DR is classified into nonproliferative diabetic retinopathy (NPDR) and proliferative diabetic retinopathy (PDR) stages according to the presence of visible ophthalmological changes and the manifestation of retinal neovascularization [13,14]. NPDR is usually asymptomatic except when associated with macular edema; however, cases with uncontrolled DM or where retinopathy’s progression is not monitored tend to progress to PDR, which is generally linked to complications that could lead to impaired visual acuity [15]. The first clinical sign of DR is the presence of microaneurysms in the retina during the mild version of the disease. In moderate diabetic retinopathy, exudates, hemorrhages, and minimal intraretinal microvascular abnormalities appear and may increase their proportion in severe stages [8]. Retinal detachment, neovascularization, along with fibrovascular tissue proliferation are features of PDR where newly formed vessels are leaky, fragile, misdirected, and the contraction of the aging vitreous cause them to rupture or if a greater force is created, it can lead to tractional retinal detachment resulting in acute or gradual loss of vision [2,9].

## 3. Role of Oxidants and Antioxidants in the Eye with Diabetic Retinopathy

The eye is an organ exposed to multiple exogenous factors, which are potentially precipitants of injury, including visible light, ultraviolet light, environmental toxins, and ionizing radiation, as well as the endogenous stress induced by the mitochondria within the eye tissues during the physiological functions of the eye [16]. This endogenous and exogenous stress produces an imbalance between oxidants and antioxidants, generating unstable reactive oxygen species (ROS) characterized by one or two unpaired electrons within their outer orbit [17]. ROS generation is normally correlated with cellular metabolic rate. The ocular surface produces lipids, aqueous, and mucin, all together form the tear film, which serves to protect and maintain the health of the ocular surface when it spreads over the eyelids [18]. The aqueous layer is produced by the tear and accessory glands. Mucins are secreted by corneal epithelial cells and conjunctival goblet cells. The lipids are secreted by the Meibomian glands located in the eyelids [18]. The ocular surface is further compromised in those patients with more severe and longer duration of diabetes disease, including those with higher A1c values, or retinopathy [19,20]. That is why the eyeball, as well as being exposed to attacks by ROS, is also provided with different antioxidants in its different segments, especially in the tear film, aqueous humor and vitreous humor.

### 3.1. Oxidative Stress and the Damage It Causes to the Eye

Oxidative stress (OS) is known as the interruption in free radical production homeostasis during various vital processes, such as the electron transport chain reaction and the sweeping of the oxidant products, or defense mechanisms designed to neutralize these harmful molecules. This imbalance is closely related to the pathophysiology of DR [21]. The addition of an electron to the dioxygen molecule creates the superoxide anion radical, which is generated mainly during the mitochondrial respiration process. Subsequently, the dismutation of this free radical by superoxide dismutase enzymes forms hydrogen peroxide (H_2_O_2_) [22]; decomposition of this molecule by various transition metals through the Fenton reaction can generate a high reactive hydroxyl radical [23,24]. Moreover, the reaction of the superoxide or hydroxyl radical with polyunsaturated fatty acids can generate the peroxyl radical (Figure 1). The human eye is constantly subject to OS, due to frequent exposure to light, in addition to high metabolic activity and oxygen tension. Solar ultraviolet radiation (UVR) turns out to be the main inducer in the external environment for ROS formation in the eye [24]. Other mechanisms like increased vascular endothelial growth factor (VEGF) production, alteration of the extracellular matrix architecture, genetic factors, and redox signaling are also present along with angiogenesis, collateral vessel formation, and increased permeability in PDR [25,26,27]. Stress induced by oxygen-derived free radicals such as hydroxyl radical, superoxide anion, and hydrogen peroxide can be harmful to cells [28], due to its ability to diffuse across hydrophobic membranes and their participation in the production of more reactive species, being H_2_O_2_ the most extensively studied oxygen metabolite [29].

In the anterior segment of the eye, H_2_O_2_ is present in the uvea and in the aqueous humor of mammals at concentrations between 30 and 70 µM [30]. H_2_O_2_ is a product of many antioxidant reactions of ascorbate such as those with oxygen and superoxide. High concentrations of this oxidant have been shown to be toxic to the lens [29,31]. Elevated levels up to seven times the normal range of H_2_O_2_ have been demonstrated in the aqueous humor and the lens of some human patients with cataracts [32]. H_2_O_2_ injected into the anterior chamber of the eye caused significant morphological changes in the iris and ciliary body and decreased intraocular pressure (IOP) [29].

In the posterior segment, H_2_O_2_ has been associated with tissue damage in the retina due to light and oxygen [33]. One of the main causes of DR is the development of glycosylated proteins, which generate free radicals, resulting in oxidative tissue damage and subsequent glutathione (GSH) depletion [34]. Glycosylated proteins can even combine with lipids and be further damaged by free radicals, forming advanced glycated end products (AGE), which can then deposit in blood vessels of the retina and promote neovascularization [35]. Diabetics with retinopathy have higher levels of oxidative damage markers in subretinal fluid when compared to diabetics without retinopathy and healthy controls [36]. The retina responds to OS with reactive gliosis: the activation of astrocytes, microglial, and macroglial cells. Microglial cells are resident retinal macrophages that confer neuroprotection against ROS damage and other injuries. OS promotes the degradation of sialic acid residues in membrane proteins, leaving photoreceptors and other cells with a damaged glycocalyx, this leads to greater phagocytosis by microglial cells, and increases neuronal cells death, worsening the pathology [37,38].

Free radicals have also been reported to cause lipid peroxidation and a decrease in potassium-evoked dopamine release in vitro. OS induced by H_2_O_2_ has shown to enhance basal release of [3H] d-aspartate but decreased potassium (K+)—evoked release of this amino acid [39]. In one study, H_2_O_2_ caused a decrease in concentrations of glutamate and glycine in the retina. While low concentrations of H_2_O_2_ also produced decrease in glycine concentration in the vitreous humor, but had no significant action on glutamate levels [40]. The catalase 3-AT inhibitor caused reduction in both the retina and vitreous humor of the glutamate and glycine concentrations, indicating an important role of endogenously produced peroxides in the regulation of retinal amino acid neurotransmission. The observed inhibitory action of H_2_O_2_ on glutamate concentrations in ex vivo experiments emulate the effects observed in in vitro assays [40]. Nitric oxide synthase—the enzyme that catalyzes the formation of nitric oxide (NO)—is located in retinal neurons and the pigment epithelium. NO can participate in reactions with superoxide radicals to form the more potent and long-lived oxidant, peroxynitrite, from which there is evidence that can inhibit the absorption of glutamate in the rat’s brain [41]. H_2_O_2_ interacts with the COX pathway that leads to the formation of prostanoids both in vitro and in vivo as well as simulating the biosynthesis of PGE*2* and PGI_2_ and the production of thromboxane B2. Isoprostanes are compounds derived from the free radical catalyzed peroxidation of arachidonic acid independent of COX. Then products such as PGE_2_ and 8-*iso*-PGF_2α_, regulate H_2_O_2_ and its inhibitory action on glutaminergic transmission in the isolated bovine retina [42]. It can be concluded that in the posterior segment H_2_O_2_ has the ability to alter the availability of amino acids in bovine eyes [29].

### 3.2. Antioxidants Present in the Tear Film, Aqueous Humor, and Vitreous Humor

The eye is packed with a variety of antioxidants, which mitigates the damaging effects of ROS. An antioxidant is frequently defined as the substance that, when present in low concentration compared to that of an oxidizable substrate, significantly delays or inhibits the oxidation of the substrate [43]. Overproduction or inadequate elimination of ROS beyond the ability to counteract the antioxidant system can cause OS and overload the eye tissues [44]. Tear film and aqueous humor are important components of defense mechanisms on the ocular surface.

The tear film covers the anterior surface of the cornea and is the first line of defense against external aggressions [24]; it contains both non-enzymatic and enzymatic antioxidants. In human tears, ascorbic acid (665 µM) and uric acid (328 µM) represent ~50% of the total antioxidant activity, with ascorbic acid being the most abundant followed by uric acid; some other small molecules found are GSH (107 µM), L-cysteine (48 µM), and L-tyrosine (45 µM). The only antioxidant enzyme reported in the tear film is superoxide dismutase (SOD), which has an activity at 1–32 U/mg protein [24,45].

Aqueous humor is a clear, slightly alkaline liquid that occupies the space between the cornea and the lens, formed and secreted by the ciliary bodies. It plays a crucial role in the nutrition and protection of the corneal endothelium and in the anterior epithelial lining of the lens. Another of its functions is to eliminate metabolic waste and biochemical products generated by the cornea and the lens. ROS can be continuously generated in the aqueous humor in the form of hydrogen peroxide, superoxide anion, singlet oxygen, and peroxyl radicals [46]. The antioxidants found in the aqueous humor are almost the same as in the tear film, among the non-enzymatic antioxidants there is ascorbic acid (530 µM), L-tyrosine (78 µM), uric acid (43 µM), L-cysteine (14.3 µM), and glutathione (5.5 µM) [45]. Ascorbic acid has three different protective mechanisms in the aqueous humor: quenching or blocking the fluorescence of biomolecules, control of the biotransformation generated by the same fluorescence, and the direct absorption of UVR. The amino acid L-tyrosine is electrochemically active and removes hydroxyl radicals and singlet oxygen species. Uric acid (UA), a water-soluble molecule with high reactivity towards singlet oxygen and hydroxyl radicals, serves as a powerful scavenger of ROS [24,47]. In summary, the tear film and aqueous humor are packed with low-molecular weight, water-soluble antioxidants, which support the cornea’s defense mechanisms against OS [24].

The vitreous humor is the structure that fills the space within the posterior segment of the eye; it is surrounded by the surface of the posterior lens and by the internal limiting membrane (ILM) of the retina. The vitreous body has a total volume of approximately 4 mL, mainly composed of water (98–99%), collagen fibers, glycosaminoglycans, non-collagen proteins, and small amounts of trace elements [48]. The nature of the vitreous gel is attributed to the interaction between its two main components: collagen and hyaluronan (HA) [44]. The concentration of HA within the vitreous gel varies between 0.02 and 1 mg/cm3 and plays a synergistic role with collagen and other proteoglycans for the regulation of vitreous stiffness [44,49]. The vitreous cortex is a lamellar structure attached to the ILM of the retina posterior to the peripheral vitreous base by an extracellular matrix “adhesive” consisting of laminin, opticin, fibronectin, chondroitin sulfate, and heparan sulfate. It can be said that the vitreous is acellular since it only presents a monolayer of mononuclear phagocytes, hyalocytes, located within the posterior vitreous cortex [50]. Among its functions, the vitreous contributes to the clarity of the intraocular media, the maintenance of IOP, and the regulation of intraocular oxygen tension [51]. In addition, the vitreous body provides protection by acting as a shock absorber, due to the collagen fibers that reduce the compressive forces of HA when the globe is exposed to external pressure [52].

The vitreous accumulates a high amount of water-soluble antioxidants, which could protect the eye from OS. These antioxidants can also be classified into enzymatic and non-enzymatic antioxidants [29]. Non-enzymatic antioxidants have the ability to quickly inactivate radicals and oxidants. Considering the source of non-enzymatic vitreous antioxidants, these can be classified into nutrient non-enzymatic and metabolic antioxidants [53]. Nutrient non-enzymatic antioxidants include those obtained exogenously through food and supplements, such as vitamin C, vitamin B2, and trace metals like zinc and selenium. Metabolic antioxidants are endogenous antioxidants produced by the body itself, such as GSH, metal-chelating proteins like transferrin, and uric acid [44,54].

Vitamin C, also known as ascorbic acid, is a water-soluble molecule present in most tissues in its anionic state but cannot be synthesized by humans: they can only obtain it exogenously. The vitreous gel receives its vitamin C supply from the plasma through active transport from the ciliary body. The ascorbic acid found inside the vitreous body reaches concentrations of approximately 2 mmol/L; this is 33 times more than the plasma concentration. As an antioxidant, ascorbic acid is oxidized to convert superoxide anions and lipid hydroperoxidases into stable forms, thereby preventing lipid peroxidation [55]. Vitamin B2 (riboflavin) has been detected in the human vitreous (0.8 µg/100 mL) and animal (8.0 µg/L). Riboflavin protects against lipid peroxidation and plays an essential role in the glutathione redox cycle [56]. Zinc is the most abundant trace element within the eye that exerts antioxidant effects by protecting sulfhydryl groups from oxidation; its concentration is close to 1.95 µMol/L. Zinc works as a scavenger of free oxygen radicals like hydroxyl as it acts as a stimulus for metallothionein synthesis, it also protects tissues from various forms of oxidative damage, including lipid peroxidation and glycoxidation [57,58]. Selenium is an essential trace element, it has an average concentration of 0.1035 µMol/L. Selenium works indirectly as an antioxidant being incorporated into antioxidant enzymes, such as selenoenzymes [59].

The GSH peptide has cysteine and a thiol antioxidant in its constitution, and is found in an average concentration of 0.26 mmol/L [60]. As an antioxidant, glutathione can directly remove selected oxygen radicals and indirectly aid in the recycling of vitamins C and E [61], it also functions as a cofactor for glutathione peroxidase (GPx) activity allowing the reduction of lipid hydroperoxides [62]. Transferrin (molecular weight ~80 kDa), is found in an average concentration in the vitreous of 0.0878 g/L [44]. As an antioxidant, transferrin is an iron chelator that keeps ionic iron sequestered at physiological pH, and thus minimizes the participation of iron in radical iron-dependent reactions; its activity helps reduce the toxicity of intravitreal iron during vitreous hemorrhage [63]. Uric acid is a breakdown product of purine nucleotides and works as an antioxidant at normal concentrations. However, in the presence of oxidative stress, there is an upregulation of UA concentrations and a change related to redox balance, causing UA to become oxidative [64].

The enzymatic antioxidants detected in the vitreous are GPx, SOD, and catalase. From the glutathione peroxidase family, extracellular GPx3 and phospholipid GPx4 are found within the vitreous body [65]. As a homotetrameric protein, GPx3 catalyzes the reduction of organic hydroperoxides and H_2_O_2_ until alcohol and water are obtained by using GSH as an electron donor. GPx4 is a monomeric protein capable of directly reducing phospholipid and cholesterol hydroperoxides [66]. SOD is a metalloprotein enzyme that is responsible for catalyzing superoxide radicals into hydrogen peroxide and molecular oxygen. SOD is made up of three isoforms: cytosolic SOD (SOD1), mitochondrial SOD (SOD2), and extracellular SOD (SOD3). SOD1 and SOD3 contain copper and zinc (Cu/Zn-SOD), while SOD2 contains manganese (Mn-SOD) [44,67]. In the vitreous base and cortex we find the concentrated SOD3 isoenzyme where it interacts with proteoglycans and regulates the response to OS in the vitreous preventing local oxidative damage [67]. Catalase is a tetrahedral hemoprotein that also protects tissues from the toxic effects of peroxide by converting peroxides to water and oxygen. The vitreous body in humans has an average concentration of 58 µL O_2_ per mg of soluble catalase protein. It has been detected in the vitreous of patients with PDR; this suggests that catalase may be a potential target for the treatment of acute ischemic diseases of the retina [68]. Besides, along with GPx are found in other ocular tissues, including the iris and the ciliary body [30]. Fluorometric and postmortem toxicological analysis studies have shown that the passage of molecules from the systemic circulation to the vitreous through the blood–aqueous and blood–retina barriers is mediated by diffusion, hydrostatic and osmotic pressure gradients, convection, and active transport [69,70]. Repeated long-term administration of these agents may be necessary to achieve sufficient therapeutic doses of exogenous nutrients within the vitreous [71].

## 4. Ocular Matrices: Tears, Aqueous Humor, and Vitreous Humor

There are many different microenvironments in the body; each organ and tissue can have its own microenvironment, including blood and cells. For example, a given biomarker can be present at multiple sites, and its relationship to the state of retinopathy can vary according to the site where it is measured [72]. The eye is a complex sensory organ that has the ability to receive light and convert it into electrical impulses, which are transmitted to the brain through the optic nerve, resulting in visual perception. In the case of animal models with ocular disorders, the variety of ocular matrices that can be collected and analyzed for biomarker measurement is wide, but the implementation of this biomarker measurement in the clinic together with the type of ocular matrix to be sampled is a key consideration. In humans, the most easily obtained eye matrices are tears and tissues of the ocular surface, such as the cornea and conjunctiva, they provide exact information regarding disorders in the anterior segment. Aqueous humor (AH) and vitreous are the most suitable matrices for evaluating relevant biomarkers for posterior segment disorders, such as DR. They are difficult to access matrices, requiring an invasive procedure performed in the clinic to facilitate specimen collection, leading to significant ocular complications [73]. Evaluation of ocular biomarkers provides valuable information regarding disease progression and this makes it a critical component of the discovery and development of ophthalmic drugs.

### 4.1. Tears

The tear film is composed by three layers: mucin, aqueous, and lipid that provides functional, nutritional, and protective characteristics for the ocular surface. These layers interact with the meibomian gland, lacrimal gland, conjunctiva, and cornea, which facilitate and regulate the normal production, distribution, and elimination of tears. The tear film creates a refractive surface on the cornea and protects it from environmental damage, each layer contents glycoproteins, amino acids, electrolytes, enzymes and proteins like lipocalin, lysozymes, lactoferrin, and albumin, as well phosphatidylcholine and phosphatidylethanolamine. Particular immunological components include sIgA e IgG, cytokines mainly Tumor necrosis factor-α (TNF-α), IL-α, IL-1ß, IL-6, matrix metalloproteinases (MMP), and chemokines, which are immunological mediators in the ocular surface diseases (Figure 2). Although, they are expressed in other diseases, for example, in autoimmune diseases, diabetic retinopathy, dry eye syndrome, ocular allergies, neurological disorders like Parkinson’s disease, and Multiple Sclerosis [74]. Obtaining the tear film requires minimally invasive procedures, making it an accessible matrix that needs to be further studied and used as a starting point for the study of a variety of eye diseases. The determination of a wide variety of cytokines, enzymes and metabolic residues in tears could be useful for the establishment of therapeutics targets, contributing to the improvement of treatments and facilitating their diagnosis.

The number of components in tear samples varies according to the technique used for their collection. Due to the small volume of tear sample (5–10 µL), two techniques are commonly used for their collection:Direct or aspiration: the collection is through microcapillary tubes (MCT) or micropipettes, the tip of the tube is placed in the cul-de-sac for 5 min, in non-stimulated tears (NST), until it forms a lake of tear, then for capillarity tears are absorbed and the sample (5.5–6.5 µL) is transferred immediately into a sterile tube with storage solution or buffer assay to produce a dilution 1:10 and storage at −80 °C. The main advantage is the amount of proteins and biomarkers obtained directly of ocular surface. A disadvantage is the loss of proteins may occur due to incomplete pouring into microvials [75,76].Indirect methods: These methods collect the samples of tears through absorbing papers like cellulose sponges or Schirmer test strips (STS), both are invasive techniques. The cellulose sponges are used frequently to analyze inflammatory markers like interleukins and MMP-9 and they are measured by enzyme-linked immunosorbent assay (ELISA) or Luminex technology using tears that were collected with Merocel, Pro-ophta, or Weck-Cel sponges. However, comparative studies with simultaneous measurements of cytokines have shown that Merocel is useful for clinical assess for cytokines/chemokines levels but have the limitation with measures of IL-7 and IL-4 due to protein stability problems with the extraction buffer [75,76,77]. In the case of STS, it may also be used for cytokine analysis assays. For collection of tears, the strip is placed on the inferior fornix of the eye and the patient should close their eyes for 5 min. After completed the time, the patient should open their eyes for remove carefully the strip and then it is collocated into a sterile 1.5 mL tube. Immediately transfer the tube to the laboratory to process later in a bead based multiplex assay or store at −80 °C. With this method the sample contains higher amounts of cellular proteins, lipids, and mucous compared with MCT and the analysis with multiplex provide high sensitivity for analyzing cytokines and other proteins [78].

### 4.2. Aqueous Humor

AH is primarily composed of water (99.9%) and small amounts of carbohydrates, vitamins, proteins, and other nutrients, as well as growth factors and cytokines. It is responsible for maintaining eye pressure among other things for the support of ocular health. It is important to know that AH is drain from the eye through one of two passive pathways: the traditional trabecular meshwork (TM) pathway and the unconventional one, which is the uveoscleral pathway [79,80].

The samples of AH are collected by aqueous route through a paracenthesis in patients undergoing cataract surgery, trabeculectomy, phacoemulsification, or post-mortem eyes. Collection volumes are small and range from 100 to 150 µL [81,82]. After aqueous humor samples are obtained they are usually stored undiluted directly at −80 °C; they can be aliquoted according to the volume obtained and no more than one freeze–thaw cycle is recommended prior to analysis [83]. Potentially valuable information can be obtained from the analysis of these samples, however, there are potential risks associated with their collection. The methods are highly invasive and can create a risk of additional damage to the cornea and lens [73]. Post-mortem eye samples will have different profiles than those collected from living patients due to the accumulation of metabolic waste and other uncontrolled post-mortem processes. This is why samples collected from living patients are considered more useful.

### 4.3. Vitreous Humor

The vitreous is located in the posterior segment between the lens and the retina. As stated earlier, it is mainly composed of water and a mesh of fine collagen fibrils embedded with dissolved hyaluronan molecules, inorganic salts and lipids [84]. In addition, the vitreous contains other proteins such as albumin, globulins, clotting proteins, and complement factors that have accumulated from blood filtration or the spread of surrounding tissue and vasculature [85]. The anatomical position of this matrix in contact and close with the retina makes it an ideal compartment for taking samples that reflect biochemical and pathophysiological changes when there are states of retinal or vitreoretinal disease, including PDR [73,86].

Vitreous samples in vivo are generally collected through vitreous taps from vitrectomy patients [87]. Samples obtained from vitreous humor (900–920 uL) are generally stored undiluted at −80 °C as well as aqueous humor, it is recommended to separate them in aliquots to avoid freeze–thaw cycles. For its analysis, the samples can be centrifuged to eliminate cellular components and debris [88]. Given the invasiveness of vitreous sampling, recently has been evaluated the possibility of collecting vitreous reflux after intravitreal injections, being the appropriate time and without the need to enter the operating room, however, the possible risks should be better assessed [89]. In a study by Srividya and colleagues, they used Schirmer’s tear strips to collect vitreous reflux from patients with diabetic macular edema (DME) and PDR, where they compared their total protein concentration with undiluted vitrectomy specimens. The results showed similar total protein concentrations between the vitreous reflux and the vitrectomy samples (*P* < 0.05) [89]. The most common method of analysis for biomarkers in this samples are enzyme-linked immunosorbent assays (ELISAs) and enzyme-colorimetric assays. Recently, multiplex bead array assays are now commonly used to maximize the usefulness of the vitreous sample, like in AH because they have the advantage that they can measure multiple analytes simultaneously in small volumes [90]. Proteomic and genomic analysis techniques are also used to analyze biomarkers of different nature in vitreous samples.

## 5. Measurement of Biomarkers

A biomarker provide a powerful and dynamic approach to improve our understanding of the mechanisms underlying eye disease, supporting information for diagnosis, disease control, or to predict clinical response to treatment [73]. Biomarkers can help to understand DR and contribute to the development of new treatments or new clinical strategies to prevent vision loss [91]. The FDA-NIH Joint Leadership Council updated the BEST Resource (Biomarkers, Endpoints, and Other Tools), modify the original definition of biomarker to “a defined characteristic that is measured as an indicator of normal biological processes, pathogenic processes, or responses to an exposure or intervention, including therapeutic interventions” [92]. New biomarkers are generally considered of interest, as they are associated to predict the disease or its response to treatments, however they are not proven, nor are they widely accepted or used in clinical practice as a decision-making tool [91]. The development of new biomarkers is a complex process that requires high refinement, and depends on the etiological disclosure that is presented. On the other hand, in clinical practice individual biomarkers but also a set or panel of biomarkers can be used, these can include biomarkers of different types, such as exposure, effect, and susceptibility biomarkers. Multiplex immunoassays are ideal for analyzing small volumes of samples, such as tears and now also in AH where they have been shown to be effective in measuring cytokine levels. With Next Generation Sequencing Techniques (NGS), small-sample miRNome analysis can be performed that avoids some of the limitations of hybridization-based detection methods [93]. Genetic matrix analysis is an additional powerful technique for comparing gene expression profiles in AH [94].

### 5.1. Oxidative Stress: ROS and Relevance in DR

The ocular surface is always subjected to intense exposure to light and solar ultraviolet radiation, also high metabolic activity increases the production of ROS and OS. This is a common adaptation secondary to inflammation and diabetes that produce more mediators to ROS in ocular surface as xanthine oxidase. It is hypothesized that the increase in antioxidant markers may reflect a local, compensatory response in the eye against OS [73]. Therefore, adequate levels of antioxidant enzymes responsible for free radical scavenging are essential for oxidation and reduction (redox) homeostasis [95]. Nezzar et al. analyzed the expression of glutathione peroxidase (GPx1–GPx8), catalase and SOD1 in human tissues of meibomian glands and conjunctiva and they found that both tissues express GPx2, GPx4, and GPx7 to control peroxide concentration in the lipid layer to protects the ocular surface from OS damage. Moreover, SOD1 and catalase were found expressed in conjunctiva and meibomian gland this might suggest their role in recycling ROS and their interaction with inflammation caused by OS [96]. In addition, products such as GPx, SOD and malondialdehyde (MDA) have been observed at irregular levels in the AH of patients with PDR. SOD is a key antioxidant enzyme involved in the metabolism of oxygen free radicals and prevents the formation of other ROS [97].

GPXs are a family of selenium-dependent isoenzymes that catalyze the reduction of H_2_O_2_ or organic hydroperoxides in water and alcohols through oxidation of GSH to GSSG. Its activity can be measured using cumene hydroperoxide and GSH as substrates in a coupled reaction with glutathione reductase (GR) [98,99]. GPx messenger RNA express mainly in the ocular ciliary epithelium suggests that in AH glutathione peroxidase originates from these cells [100], its activity in the AH of glaucoma patients has been reported to be three times higher than in cataract patients, the mean value of GPx in AH of glaucoma patients was 18.4 ± 2.5 U/mL, and the mean value of the cataract group was 6.1 ± 0.6 U/mL (*p* < 0.001) [101] (see Table 1). Some antioxidant mechanisms, such as glutathione-related enzymes, are likely to work in normal and vitreous lenses to keep proteins in a reduced state. Unstable pro-oxidant molecules in the vitreous can play a role in cataract pathogenesis along with retinal disease. They can spread damage to lens membranes and vitreous proteins [102,103]. The study by Altomare et al. said specific activity of the GPx could not be detected in the vitreous humor; it is probably due to the high dilution of the vitreous humor used for the extraction procedure [103].

SODs are a family of enzymes that catalyze the dismutation of superoxide into oxygen and H_2_O_2_ [99,131]. The fluid in the anterior chamber has very low SOD activity, so there are no apparent differences between diabetic patients and controls. However, it should be noted that the humor of the anterior chamber has a high content of ascorbate, which contributes significantly to protection against superoxide [132]. Significantly lower SOD levels in the vitreous have been found in diabetic patients with PDR than in non-diabetic controls examined post mortem [132]. There is a flow of fluid from the vitreous to the choriocapillary that holds the retina together, this is where SOD can contribute to protection against superoxide radicals in the extracellular space of the retina [132].

Determination of total antioxidant capacity (TAC) is a method for rapid quantification of antioxidant efficacy in disease prevention [133,134], defined as the “cumulative action of all antioxidants present in plasma and body fluids, providing an integrated parameter rather than the simple sum of measurable antioxidant values” [135]. Reduced capacity has been observed in the AH of patients with PDR [104] and there is a close correlation between OS and morphological changes in the TM [136], in patients with PDR suggests that the involvement of the anterior chamber may be caused in part by redox state imbalances [104]. Actually, the antioxidants of each system can exert their activity with different mechanisms and different efficiency; therefore, the assessment of TAC could be much more important than the concentration of antioxidants individually [104]. Izuta et al. reported that the TAC scan of the vitreous actually increases in patients with PDR [137].

Oxidation of DNA components by ROS and reactive nitrogen species (RNS) is the main source of DNA damage leading to various types of DNA modification, including chain breakage, nucleotide oxidation, base loss, and adduct formation [138]. The radicals can react with all the purine and pyrimidine bases, as well as with the deoxyribose skeleton, generating several products, one of them 8-hidroxi-2-deoxiguanosine (8-OHdG) [139]. It has been estimated that several thousand 8-OHdG lesions can form daily in a mammalian cell, this represents 5% of all oxidative lesions, which is why it is one of the most widely used DNA oxidation biomarkers to measure OS [140]. Their levels in aqueous humor increases and the total antioxidant status decreases in the serum and aqueous humor of glaucoma patients [141]. High myopia is a degenerative disease [142]. Therefore, 8-OHdG was found at lower levels in it compared to the control group; their levels are positively correlated with central corneal thickness (CCT) and negatively correlated with long axial length (AXL). Myopic patients presented 212.5 ± 103.2 µg/mL versus 311.6 ± 127.7 µg/mL in the control group with cataract patients [109] (see Table 1).

In diabetic patients with PDR, lipid peroxidation is a very pronounced process in humoral parts of the body, which is why it is responsible for the OS induced in diabetic patients [143]. Polyunsaturated fatty acid (PUFA) molecules are present in cell membranes and are prone to oxidation due to the presence of susceptible carbon double bonds [144,145]. The eye, in general, but particularly the macula, is susceptible to OS due to its high metabolic activity and the large amount of PUFA in the photoreceptor membranes [145]. The determination of the final product of lipid oxidation is widely used as a marker of OS. MDA together with 4-hydroxy-2-nonenal (4-HNE) represent the most investigated final product of lipid oxidation. 4-HNE is a stable and biologically very active molecule whose presence is not restricted to the site of its origin, since it can diffuse through the membrane barrier [146]. The elevated blood levels of MDA in patients with diabetes reflect similar increases in the level of AH, which points to the involvement of OS and lipid peroxidation in the progression of DR to the proliferative form [95]. Cell-free vitreous is the target of 4-HNE, although aging has been reported to increase specific conjugates in retinal proteins [147]. After laser treatment, this biomarker accumulates in this relatively separate compartment giving rise to conjugated humor proteins and deterioration of the antioxidant activity by the lipoperoxidation product will make the vitreous more vulnerable to pro-oxidative effect [148]. Besides, it could interfere with the expression of different collagen subtypes effecting a change in the expression profile of the collagen subclasses, having consequences for the functional integrity of the vitreous as seen in the aged eye [149]. On the other hand, the results on the changes of MDA in the vitreous of diabetic patients with PDR obtained by Brzović-Šarić et al. agree with what has been reported, which showed an increase in vitreous MDA in patients with PDR. So in general, direct lipid peroxidation (LPO) method is more sensitive and provides a better picture of the status than MDA [105]. A significant correlation has been shown between the increase in LPO in the vitreous and an increase in the expression of VEGF; therefore, it seems that the determination of LPO in serum could be a good predictor of the onset of OS in the vitreous [105,150]. Because the retina is rich in PUFAs and there is an increase in glucose oxidation and oxygen absorption, it is susceptible to an increase in OS. The structural and functional changes observed in it may be due to lipid peroxidation of the vascular endothelium [151]. Isoprostanes (F2-IsoPs) are stable products, whose formation increases with exposure to OS, they have gained acceptance as a reliable marker of oxidative damage in both in vivo and in vitro animal models and their use is becoming more frequent [32], as a chemically stable prostaglandin-like isomers generated by the reaction of PUFAs in membrane phospholipids. They are formed in lipid membranes and then released freely by the action of phospholipase. An important fact is that they are not affected by lipid content in the diet and, therefore, its measurement in biological fluids can provide an estimate of endogenous production [152].

NO is one of the most abundant free radicals in the human body, which can also react with other ROS, this causes cellular dysfunction and apoptosis [153]. It is synthesized by endothelial cells and is an important vasoactive agent that affects blood flow along with other vascular functions [154]. In the eye, neuronal nitric oxide synthase is believed to be responsible for producing NO in photoreceptors and bipolar cells, and as far as endothelial nitric oxide synthase is concerned, it is present in vascular endothelial cells [155]. However, inducible nitric oxide synthase found in Muller’s cells and in the retinal pigment epithelium could be involved in phagocytosis of the external segment of the photoreceptor, in addition to infectious, inflammatory, and ischemic processes, and in the pathogenesis of DR [156]. In a Kulaksızoglu’s study, the NO levels found by the measurement of nitrite and nitrate in the aqueous humor were significantly higher in patients with PDR than in controls without diabetes [95]. Another study noted that improved NO production in the eyes of people with diabetes is consistent with reports of it as a mediator of physiological and pathological processes in the retina [157]. NO is a highly important intercellular signaling molecule that plays a role in vasodilatory responses. In addition, it is involved in basal conditions and after retinal ischemia in the control of retinal blood flow [158]. The use of aqueous humor as a biological sample is proposed to assess the course of different eye disorders showing an increase in pro-oxidative molecules and reduction in antioxidants [159,160]. In the study by Yilmaz et al. nitrite levels in vitreous humor of patients with PDR were higher than those of the control group (patients undergoing vitrectomy for idiopathic macular hole). However, among patients with PDR, there were no significant differences between the levels corresponding to type I and type II diabetes (*P* = 0.56). On the other hand, patients with type I diabetes had a mean concentration of 0.524 ± 0.27 µM in vitreous, while those with type II diabetes had a mean of 0.383 ± 0.17 µM [112] (see Table 1). Other possibility is that overproduction of NO can cause damage to the retina by disrupting the rod outer segment membrane phagocytosis by retinal pigment epithelium (RPE) cells. Then, an accumulation of ROS occurs between the photoreceptors and the RPE cells, resulting in the degeneration of the photoreceptors [161].

### 5.2. Inflammation: Cytokines and Relevance in DR

DM patients with advanced diabetic retinopathy have more dry eye issues associated with neurotrophic keratopathy which could cause severe vision loss due to cornea ulcer or neurotrophic ulcer, this could lead an increased expression of proinflammatory cytokines, like inducible protein-10 (IP-10) and monocyte chemoattractant protein-1 (MCP-1), and decreased levels of antiangiogenic cytokines which demonstrate that an inflammatory reaction occurs in the ocular surface of diabetic patients [162]. The alterations in the ocular surface have been associated to inflammatory processes and microvascular damage that involves mediators like Th1 and Th17 and IL-1ß, IL-17A, TNF-α, and mainly Epidermal Growth Factor (EGF) found elevated in patients with diabetes mellitus [126].

TNF-α a cytokine strongly correlated with insulin resistance by changing phosphorylation of insulin receptor substrate-1 and interfering with insulin signaling cascade and leading chronic inflammation due to leukostasis induce by VEGF, IL-1α and platelet-activating factor in the retinal vasculature, but also it is a mediator of apoptosis in retinal neurons and endothelial cells. TNF-α in tears was significantly higher in patients with PDR (13.5 pg/mL) compared with NPDR (2.8 pg/mL) [122]. In a recent cross-sectional study in Asian patients with DR, there was evaluated the differences in Total Protein Concentrations (TPC) and TNF-α in tears correlated with the three stages of NPDR (mild, moderate and severe), they found a decreased in TPC in moderate and severe NPDR compared with higher concentrations of TNF-α in the same stages, but in mild NPDR and patients without retinopathy the concentrations were similar. In this study, there was also higher levels of TNF-α correlated with higher levels of HbA1c which are explained by AGEs that activate proinflammatory pathways and promotes angiogenesis and microvascular changes in the retina [163]. TNF-α is one of the cytokines that induces disruption of the BRB by loosening the tight junctions between individual retinal endothelial cells and also between cells of the RPE, causing BRB breakdown. It promotes the irreversible adhesion of leukocytes to the endothelium and increases the production of ROS [164]. It also plays an important role in neovascularization and vasomotor reactions [165]. Actually, increased levels of TNF-α have been found in the vitreous body of patients with T2DM and PDR compared with a control group [166]. There is a study where patients with PDR were divided into subgroups based on disease progression and regression, the vitreous levels of VEGF and IL-6 were significantly higher in the eyes of patients in the progression group with 1789.2 pg/mL (198.5–3436.8) and 347.2 pg/mL (26.2–758.6), respectively (see Table 2), than they were in eyes with PDR regression [116].

In the inflammatory pathway, IL-6 has a role with induction of multiple process like synthesis and release of acute phase reactants and matrix metalloproteinase 9 (MMP-9), decreased tear production and apoptosis, induced differentiation of Th17, and the release of other proinflammatory cytokines like IL-1ß, the increased levels in this cytokine are particularly associated with increased metaplasia and keratinization of the ocular surface. Additionally, the secretion of interferon gamma (IFN-y) plays a multiple role with adaptive and innate response on the ocular surfaces inducing cell loss in the conjunctival goblet cells that reduces mucin production and apoptosis on lacrimal acini, associating high levels of this cytokine with severity of the tear film dysfunction [73,113,185]. Furthermore, decrease in the concentrations of proteins lysozyme, lactoferrin and albumin is more frequent in patients with PDR than NPDR [186], that could be associated with changes on ocular surface like tear film dysfunction, progressive loss of corneal epithelia and degeneration of nerve fibers [187], increasing the risk of corneal ulcerations and deteriorate their visual quality. In other studies have been analyze tear samples in different stages of diabetic retinopathy and found a decreased in the content of tears proteins in the onset of DR as well, tear film dysfunction due to malfunctioning in the tear formation or more diluted tears. Lactotransferrin (LTF) and Immunoglobulin λ are increased in PDR together with another tear proteins expressed with frequency in PDR like lipocalin 1 (LCN1), lysozyme C (LYZ), lipophilin A, lacritin (LACRT) lipidic carriers for retinoids required for tear production, their increase are specifically associated as predictors for DR progression, some are linked to inflammation secondary to neovascularization, bleeding or macular edema [188]. Interestingly, IL-6 levels were positively correlated with the DR stage (PDR: 47.68 vs. NPDR: 29.68 pg/mL; *p* < 0.001) [115]. Elevated levels of IL-6 have been reported to participate as a proinflammatory and angiogenic factor in PDR and DM, is also involved in crossing both the blood–brain barrier and the blood–retinal barrier [189,190], and their levels in DR patients increased significantly compared to those in a non-diabetic control group. IL-6 levels were significantly correlated with PDR and it can act as one of the main drivers to generate a change in the cytokine profile in the aqueous humor of DR patients [115,191].

AH analysis provides useful information to understand the pathogenesis and responses to treatment of various eye conditions. The analysis of AH of DR patients has allow to identify some of the mediators (cytokines, chemokines, among other factors) involved in the pathogenesis of DR. Like IL-8, the concentration of VEGF is found to be increased in AH of diabetic patients [192]. Most studies have worked on aqueous humor cytokines in patients with PDR. However, the dynamic changes between the levels of cytokines in the different states of severity of DR must be analyzed in greater detail. Chen et al. found that IL-6 had positive correlations with IL-8, IP-10, leukemia inhibitory factor (LIF) and hepatocyte growth factor (HGF) in the DR group, while it presented negative correlations with IL-9, IL-21, IL-23, IL-27, and IL-31.

IL-8 is an important chemoattractant that regulates chronic leukocytic inflammation in the vascular walls and ultimately leads to capillary occlusion and retinal ischemia [193], its levels have been reported in the AH of DR patients with 42.20 ± 33.03 pg/mL (mean ± SD) [115]. IL-8 expression can be hypoxia-induced and mediated by nuclear factor kappa B (NF-kB). IL-8 can induce angiogenesis in vitro and in vivo, and its elevated levels have been found in aqueous and vitreous of patients with ocular vascular disease, including DR [194]. Other studies have reported that elevated levels of vitreous IL-8 are associated with increasing levels of retinal ischemia and an increased degree of gliotic obliteration of large vessels in patients with PDR (see Table 2) [119,193]. Sun et al. found that the level of IL-8 in aqueous humor was significantly higher than that in the vitreous in PDR patients [119].

IL-10 is an anti-inflammatory cytokine and antiangiogenic mediator produced by monocytes and macrophages its antitumor effects have been associated with its ability to prevent angiogenesis associated with tumor growth [195]. In Mao C study, the median IL-10 concentration was higher in the vitreous of the PDR patients (224.789 pg/mL) than in the control group (160.143 pg/mL) and it was statistically significantly, which was different from Hernández C study [121,196]. A predominance of macrophages (50%) has been determined in the vitreous samples of patients with PDR by cytological examination, this could explain the greater production of IL-10 resistant to other proinflammatory cytokines due to the increase in macrophages [125]. It suppresses the expression of the receptor for proinflammatory cytokines such as IL-1 and tumor necrosis factor α (TNF-α) and inhibits the activation of its receptors by reducing the synthesis of these cytokines it limits inflammation. Its antiangiogenic effect has recently been shown to be associated with the downregulation of VEGF expression [197]. Dong et al. found that IL-10 levels in the AH decreased with increasing severity of DR.

TNF-α is one of the main inflammatory response cytokines with chemotactic action on monocytes and neutrophils that activates them as macrophages, improving their cytotoxicity, while being one of the mediators of this cytotoxicity [198], with levels in aqueous humor of 4.04 ± 1.83 pg/mL (mean ± SD) in DR patients [115]. VEGF, also known as vascular permeability factor (VPF or vasculotropin), considered today the main factor controlling angiogenesis and vascular permeability and causes much of the pathogenesis of PDR and diabetic macular edema [199]. It is produced by endothelial cells, macrophages, CD4 lymphocytes, plasma cells, among others [200,201]. One study showed that the VEGF level was significantly elevated in diabetic subjects compared to the non-diabetic control group. However, there was no correlation between VEGF and DR severity (*p* = 0.357) [115].

The vitreous cavity contains different types of unique cells, such as retinal cells, RPE, choroid, and retinal vessels. Although the AH and the vitreous fluid do not flow with each other, some proteins can be exchanged between these two compartments. IL-6, IL-8, VEGF, and TNF-α have been observed to be elevated in serum and vitreous fluid from patients with DR [202]. Murugeswari et al. found that the levels of IL-6, IL-8, MCP-1, and VEGF present in the vitreous humor were significantly higher in PDR patients than in patients with macular hole [203]. In another study, a comprehensive analysis of mediators in the vitreous fluids of patients with PDR and in patients with other eye diseases was performed and elevated levels of VEGF, MCP-1, IL-8 and IL-6 were found compared to the control group [204]. IL-6 normally participates in acute phase reactions such as hematopoiesis, IL-2 induction, and differentiation of keratinocytes and B and T lymphocytes. In vitro studies suggest that NO may be acting as the intermediate molecule in IL-6-induced VEGF synthesis. Precisely, the synthesis of IL-6 can be induced in vitro by hypoxia and tissue hyperglycemia [205].

Interleukin 17 (IL-17) plays an important role in a wide variety of immunological diseases, but on the ocular surface it promotes neutrophil infiltration in tissues that induce synthesis and secretion of matrix metalloproteinases and ROS that generate disruption in the corneal epithelium, loss of epithelial functionality and induction of apoptosis. This contributes to the induction of neovascularization with stimulation of proangiogenic factor and modulation of cytoskeleton, which was examined in the vitreous and plasma of patients with DR, with levels significantly increased in PDR [114,206,207].

### 5.3. Apoptosis

Apoptosis is the most studied type of cell death in diabetic retinopathy its characteristics are well defined and it is easily detected with techniques such as TUNEL (Terminal dUTP Nick End Labeling). Despite this, there are types of cell death that are difficult to detect due to the lack of defined markers and available techniques [208]. The hyperosmolar state has a relevant change in various tissues, but in tear film and cornea increases desquamation, loss of intercellular connections, disruption of cell membranes and decreases cytoplasmic density, it was found that proinflammatory stimulus in the corneal epithelium increases the expression of cytokines like IL-1ß, TNF-α, IL-8, MMP-9 activating the MAPK signaling pathway then the expression of proapoptotic markers (Fas, Fas ligand, APO2, CD40 and CD40 ligand) and cytochrome-c, this mediates an apoptotic pathway reported in vitro with an hyperosmotic state (≥450 mOsm) which converge on proteolytic activation of caspase-3 capable of cleaving various cellular proteinases to finally cause apoptotic death [130]. An increase in pericyte apoptosis in retinal tissue of diabetic patients compared to non-diabetic patients has been demonstrated with the use of TUNEL staining [208,209].

Lipocalin is a tear endonuclease which plays a role in the catalytic activity of DNA and its effect in the concentration of NaCl, Mg^2+^, Ca^2+^ and variation in pH facilitates lipocalin its role in prevention of viral infections and anti-inflammatory activity by regelation of tear viscosity, capture and release of lipids, inactivating endonucleases and pathogens binding sites on the ocular surface [210,211]. In diabetic patients with neurotrophic keratopathy, have been found increasing levels of AGE’s products in cornea which activates a signal pathway mediated by NK-kB, generating an increase on oxidative stress, and anormal accumulation of MMP-9, this is correlated with ocular surface inflammation and induce a cycle of damage and apoptosis [212,213]. Cathepsins are a group of proteases key players in extracellular space, they have collagenase and elastase activity and participate in extracellular matrix (ECM) remodeling, but also are signaling molecules. Cathepsin C and B are present in tears, but its role in the ocular surface is not well explained. Cathepsin S is involved in initiating inflammatory responses by effecting degradation of lysosomal protein and ECM which is essential for homeostasis and cellular differentiation [127].

Matrix metalloproteinases (MMPs) are a family of calcium-activated zinc-containing enzymes that are involved in turnover and remodeling of the ECM and collectively are capable of breaking down most of their protein components, including collagens, laminin, fibronectin, elastin and other components. MMPs can also degrade a number of non-ECM proteins, including growth factors, chemokines, cytokines, and some surface receptors [214]. Metalloproteinases are involved in a number of both physiological and pathological processes, the enzymes may participate in pathological processes such as neovascularization [128]. MMP-2 and MMP-9 can degrade pigment epithelial derived factor (PEDF), which is the main antiangiogenic protein of the eye, specifically it is a 50 kDa glycoprotein highly expressed in the retinal pigment epithelium [215]. We know that one of the first characteristics of DR is the rupture of the blood–retina barrier. This results in vascular permeability in the retina and the development of retinal edema. These types of findings suggest that elevated MMP expression in AH may facilitate an increase in vascular permeability [216]. MMP-9 belongs to the group of type IV collagenases which plays an important role in new vessel formation and were previously found to be upregulated by VEGF [217]. Increased levels of pro-MMP-9 and activated MMP-9 have previously been found in the vitreous of patients with PDR associated with vitreous hemorrhage [218]. MMPs can also act in the early stages of DR, in the breakdown of the BRB and in the destruction of the tight junctions of endothelial cells [216]. Among the different MMPs examined in the vitreous samples, only the levels of MMP-2 and MMP-9 levels are significantly increased in the PDR eyes. In another study both metalloproteinases are present in the vitreous samples from PDR patients, being MMP-9 the only one elevated in PDR patients [128]. Noda et al. discovered that the vitreous proliferative membranes in diabetic retinopathy, in addition to having elevated levels of MMP-9, contain high levels of MMP-2. Furthermore, elevated levels of MMP-9 have been previously reported in the vitreous humor and fibrovascular membranes of patients with PDR [219].

The release of cytochrome c (*cyt c*) from the mitochondria is a fundamental step for the beginning of the apoptotic process [220]. *Cyt c* is a small globular protein that contains iron porphyrin cofactor (heme c) that covalently binds to the unique polypeptide chain. Its main function is its participation in the electron transport chain of the inner mitochondrial membrane (IMM). *Cyt c* is reversibly reduced and oxidized as an electron is transferred from ubiquinol-cytochrome c reductase (complex III) to *cyt c* oxidase (complex IV) in the mitochondrial respiratory chain [221]. Premature death of retinal cells occurs prior to the development of other lesions characteristic of retinopathy, suggesting that it may play a critical role in the development of DR [222]. The rate of apoptosis of vascular and nonvascular retinal cells in diabetes is low, but statistically higher than normal, which is consistent with the slow development of DR [223]. *Cyt c* has the ability to perform a variety of functions depending on the site and conditions of the cell where it is located; These properties have allowed it to be identified as an “extremely multifunctional” protein (EMF) [224]. So the structural and biological properties of the complex (ferric) *Cyt c-CL* (cytochrome c—cardiolipin), promotes the transformation of proteins into a peroxidase in the early stages of cellular apoptosis [225]. Besides, *cyt c* in mammalian cells can activate caspases, a family of cysteine proteases that have within their functions that of cleaving crucial substrates to induce cell dismantling [225].

Caspase-3 activation is a slower process compared to increased OS in diabetes, or from another point of view, its activation may be occurring as a consequence of increased OS [226]. Caspases involved in the activation of proinflammatory cytokines and the initiation and execution of apoptosis [223]. Because apoptosis of retinal capillary cells probably contributes to capillary “dropout” and retinal ischemia in DR, there is an interest in caspases that may be involved in initiating and executing this apoptotic process [223]. Caspase-3 is the executioner caspase that plays a central role in the proteolytic cascade during apoptosis its immunoreactivity occurred in ganglion cells in diabetic retinas, following their apoptotic death induced by ischemia, excitotoxicity, axotomy, and chronic ocular hypertension, their inhibition reduces apoptotic cell death induced in retinal cells [227]. Nuclear factor-kβ (NF-kβ) is an important polyphenic nuclear factor which participates in apoptosis and cellular neovascularization, it can be activated by a variety of signals such as IL-1β, TNF-α, and OS. Several studies have shown that is closely related to inflammation, the appearance of tumors, cellular apoptosis, among other pathological processes. However, the main pathological changes in the DR include retinal cell apoptosis and neovascularization. This highlights that NF-kβ plays a role in DR [228,229]. In another scenario when cells are stimulated by hypoxia, hyperglycemia, and some inflammatory cytokines, the NF-kβ-specific inhibitor protein (IkB-α) is phosphorylated and degraded just to be released and activated, entering the nucleus to regulate gene transcription [229].

The BRB function of endothelial cells is supported by surrounding cells, such as Müller cells, pericytes, and astrocytes. As the blood–retinal barrier depends to a large extent on this microenvironment where the function of a specific type of cell depends on the support of other types of cells, any cellular injury or cellular loss will have great effects on the proper function of the retinal barrier and, in fact, any retinal function [230,231]. The NF-kβ present in the subretinal membranes and micro vessels is activated in response to increased ROS and AGE, which in turn further activate the apoptosis process. The activated NF-kβ is then further bound to nuclear DNA and thereby overexpresses different genes that lead to free radical production and further cell death [232]. Activated NF-kβ also increases the expression of cytokines IL-1b, IL-6, and IL-8 and the proapoptotic molecule caspase-3 in vitreous fluid and serum, leading to inflammation-mediated cellular apoptosis [233]. Together, the activation of NF-kβ, TNF-α, and interleukins improve MMP-9 transcription leading to DNA alkylation and the development of DR [234].

## 6. Metabolic Memory of Oxidative Stress in Diabetic Retinopathy

Several studies maintain that epigenetic modification is a significant factor in the development of DR. The duration of hyperglycemia decides if better subsequent glycemic control would be effective in the DR, which implies that the sustained state of hyperglycemia produces a metabolic memory phenomenon and be attributed to epigenetics, which may be the reason for inter-individual differences in drug response and variation in the progression of diabetes complications [235,236,237].

There have been several epigenetic modifications studied in DR: methylation directly to DNA molecule, which can repress transcription of certain genes; chromatin remodeling and modifications to the DNA condensing proteins, histones, which can also activate or suppress DNA on its own; and non-coding RNA that post-transcriptionally regulates gene expression. Although there are more epigenetics modifications known involved in DM or cancer, these alterations are the most studied in DR functioning as potential prognostic, therapeutic, or diagnostic biomarkers.

### 6.1. DNA Methylation

The most primitive epigenetic modification is DNA methylation and, according to various studies, it has a correlation with the progression of DR. In DNA methylation the methyl group is transferred from S-adenosylmethionine (SAM) to DNA molecules, this reaction is catalyzed by DNA methyltransferases. It was found that patients with DR have shown a significantly higher level of DNA methylation compared to those without DR, indicating that increased DNA methylation is a relevant component in the development of DR [237]. Another study revealed that methylation and activation of the matrix metalloproteinase 9 (MMP-9) gene associated with DR has an important role in accelerating apoptosis of retinal vascular endothelium [238]. This indicates that DNA methylation in DR is a highly dynamic process, which involves various epigenetic changes targeting extracellular matrix, manganese superoxide dismutase (MnSOD) also cross-linking with OS, and mitochondrial homeostasis causing mitochondrial dysfunction with subsequent capillary damage [238,239,240,241].

ROS generated from mitochondrial oxidative phosphorylation is the most representative source of OS in endothelial cells, which cause peroxidation of PUFAs, protein damage and consequently mitochondrial DNA (mtDNA) damage [239,242,243]. Methylation of mtDNA has recently been associated with the development of diseases and as a potential biomarker. Iacobazzi divided mtDNA methylation into global mtDNA methylation biomarkers, such as 5-methylcytosine (5mc) derived from the incorporation of a methyl group at position 5 of cytosine, and 5-hydroxymethylcytosine (5hmc), produced from 5mc through a hydroxymethylation reaction catalyzed by ten-eleven-translocation, which are related to aging and neurodegenerative disorders; and specific methylation biomarkers such as ND6 gene, which is suppressed under OS conditions by increased DNA methyltransferase (DNMT) [244]. As a general methylation biomarker, 5hmc can be seen at rac family small GTPase 1, which leads to increased binding of NF-kB linking this epigenetic modification with inflammation in DR [245]. In the retina, there is a more specific area where methylation is occurring called *D-loop* where transcription and replication of mtDNA is controlled [243,246] which has been used as a biomarker in a clinical study that showed significantly higher methylation rates in the serum of patients with PDR than patients without DR, these had higher methylation rates than non-diabetic patients [247] functioning as a novel diagnostic tool. Another potential biomarker related with DNA methylation in DR is homocysteine, of which high levels have been associated with increased risk of developing DR in diabetic patients, and with higher global DNA methylation probably linking it to altered metabolic memory phenomenon [248,249]. Interestingly, high homocysteine levels have been associated with increased inflammatory cytokines and activation of NF-kB [250].

Several therapeutic approaches have been assessed targeting mitochondrial dysfunction despite the low specificity of DNMT, synthetic DNA methylation inhibitors such as hydralazine and procainamide are being evaluated in clinical trials, or polyphenols like resveratrol have shown to directly inhibit DNMTs as well as being a powerful antioxidant which ultimately lead to epigenetic changes with altered gene expression [251,252,253,254].

### 6.2. Modification of Histones

Other relevant epigenetic alteration is the modification of histones, which has also been a key contributor to the pathophysiology of the DR. Histones are proteins in the nucleus related to DNA and play a role in regulation of gene expression and can be post-translational modified by methylation, acetylation, ubiquitination and phosphorylation regulating chromatin structure, and these histones can have an active cross-talk with other histone modifications [235,255]. Relevant examples of these epigenetic changes are histone acetyltransferase (HAT) and histone deacetylases are involved in regulating gene expression in the complications of diabetes [256]. For example, H3 acetylated histone expression was decreased in induced diabetic models. Furthermore, it was discovered that these changes were irreversible once the blood glucose of these rats was restored to a normal level, indicating that the development of DR is associated with the modification of histones and that it would also be participating in the phenomenon of metabolic memory [241,257]. When there is a high-glucose environment in the retina, histone acetylation is increased due to a decreased activity of histone deacetylase (HDAC) inducing an increased expression of pro-inflammatory cytokines in Müller cells [256].

During DR, an alteration of mitochondrial homeostasis and dynamics occurs, where a vicious circle is created between the alteration of mitochondrial enzymes that induce the formation of superoxide, which in turn alters the physiology of the organelles. Furthermore, dysfunction of the repair pathways generates even more mitochondrial damage [239]. However, regulation of the mtDNA replication/repair machinery has the ability to prevent mitochondrial dysfunction and the development of DR to some extent [239,258]. Decreased mitochondrial SOD2 and inhibition of Nrf2 (nuclear factor-(erythroid-derived 2-)like 2), a transcription factor affecting antioxidants, have been observed. What happens is that during the state of OS, Nrf2 translocate to the nucleus where it binds to the antioxidant response element (ARE). On the other hand, Keap1, an inhibitor of Nrf2, binds it in the cytosol and leads it to proteomic degradation through cullin-3-dependent degradation [2,252,259]. Moreover, in vitro studies have shown metabolic memory phenomenon through exposure of retinal capillary cells to high glucose concentrations, inducing epigenetic modification of SOD2; the gene that encodes for MnSOD, by decreasing methylation of H3K4 at Sod2 promoter predisposing to worsening of the oxidative stress damage cycle [255,260]. There have been different therapeutic approaches regarding histone acetylation/deacetylation in diabetic nephropathy; histone deacetylase inhibitors like Vorinostat may have a protecting effect by decreasing OS, agents like sodium butyrate inhibited HDAC activity and thus elevated the expression of Nrf2, both protecting against renal injury [256]. These therapeutic approaches have yet to be assessed in DR, but HDAC inhibition may have ameliorative effects in diabetic microvascular complications.

A study of the effects of diabetes on nuclear–mitochondrial communication in the retina revealed that the mitochondrial biogenesis of the retina is weakened in diabetes and is under the control of superoxide radicals. Therefore, this makes us think that the regulation of biogenesis by pharmaceutical or molecular means could provide a way to prevent the development/progression of DR [261]. Examples of these are approaches with intravitreal adeno associated viral MnSOD which showed to reduce retinal capillary basement membrane thickness, inhibit apoptosis of these capillaries, by effectively overexpressing MnSOD, suggesting an ameliorative effect on the metabolic memory phenomenon [254,262]. Moreover, different approaches have been made with different polyphenols, the most promising one is resveratrol that is associated with decreased phosphorylation of 5’-adenosine monophosphate activated protein kinase (AMPK) that regulates histone deacetylase Sirtuin 1 (Sirt1) which may ultimately suppress NF-kB activation [252].

### 6.3. Chromatin Remodeling

Chromatin remodeling is the conjunction of various structural and/or molecular changes that ultimately modify DNA functions as expression, replication and recombination of genes. Therefore, without having genomic alterations the production of proteins will be affected and may be inherited to offspring [258].

Histone modifiers (acetylation, deacetylation, methylation, and demethylation enzymes), histone chaperons and ATP-dependent chromatin remodelers are responsible for chromatin restructuring [263]. Four families of chromatin remodelers have been described: SWI/SNF, NO80, CHD and ISWI. This very last is associated to the Base Excision Repair (BER) and the Nucleotide Excision Repair (NER) mechanisms to help repair DNA after an oxidative insult [264]. Chromatin remodeling by its remodeler family SWI/SNF contributes to the response against stress and senescence, at the same time mitochondrial dysfunction induces chromatin remodeling responses [265]. The thioredoxing-interacting protein (TXNIP) is a protein known to have an important impact over OS, inflammation and apoptosis in the pancreas and the retina [266], in recent years it has been considered that these modifications lead to chromatin remodeling in retinal cells. During hyperglycemia some molecules such as Angiotensin II, transforming growth factor-β (TGF-β) and AGEs are overexpressed leading to enhanced activation of their receptors causing augmented production or intracellular factors that end up modifying chromatin structure [267].

The DNA binding proteins HMGBs (high mobility group B) can mediate nucleosome remodeling. Such proteins lower their affinity to DNA and their function when oxidized [268]. OS promotes alterations in chromatin by histone acetylation and increased activation of NF-kB resulting in major inflammatory genes expression [269]. Second messengers are also able to modify chromatin structure via chromatin-binding proteins. PI5P (phosphoinositide phosphatidylinositol-5-phosphate) is an example, its production augments in OS environment, it binds to ING2 (inhibitor of growth family member 2) to repress pro-proliferative genes and promote apoptosis [270,271].

### 6.4. Non-Coding RNA

Non-Coding RNAs are RNAs that are not translated into proteins, and instead they play a role as gene expression modulators. There are two types of non-coding RNAs classified by their length, small RNAs formed by 20–30 nucleotides and LncRNAs (long non-coding RNAs) having up to 200 nucleotides [272]. Small non-coding RNAs (20–30 nucleotides) are categorized as microRNAs (miRNAs), short interfering RNAs (siRNAs), and piwi-interacting RNAs (piRNAs). The first two are more similar between each other as they both have double-stranded precursors, while piRNAs appear to derive from single-stranded precursors [273].

MiRNAs have a regulatory function on endogenous genes, and siRNAs protect the genome from invasive nucleic acids [273]. Both molecules work primarily by silencing genes recognized by a specific sequence in the miRNA or siRNA chain. Normally, miRNAs inhibit the expression of their target genes promoting its mRNAs degradation or inhibition of its translation [274], but this programmed action may be modified in response to external threats (viruses, transposons) or changes within the cell. Then, siRNAs will co-opt the invader into their own mechanism and prevent it from its expression; on the other hand, miRNAs may be diluted or exchanged by different miRNAs or even new miRNAs will be expressed to silence genes that will have the effect of counteracting the previous silencing [273,275]. Most lncRNAs are transcribed by RNA polymerase II and spliced into various isoforms which act as epigenetic regulators. They can regulate DNA polymerase activity, around 20% of lncRNAs recruit chromatin remodeling complexes to repress transcription of target genes [276,277].

Through the years non-coding RNAs have been related to many degenerative illnesses, including Alzheimer, cancer and diabetes [272,274,277]. As mentioned before, when cells encounter environmental modifications, such as hyperglycemia, they promote altered expression of miRNAs. The downregulation of certain miRNAs (miR-126 and miR-200b) and the upregulation of others (miR-18a, miR-20b, miR31 an mir-155) are now related to augmented VEGF production [278], miR-146 is a miRNA that functions as a negative feedback to NF-kB activation induced by IL-1β in retinal endothelial cells and it is also upregulated after 3 months following the onset of diabetes [278].

Some miRNas such as miR27b respond to stress by reducing the expression of pro-antioxidant proteins as Nrf2 resulting in a pro-oxidant environment [279]. mR27b also has a pro-angiogenic effect targeting Notch ligand D114, Sprouty -2, PPARgamma and Semaphorin 6A [280]. Another miRNA; miR-211 is upregulated in diabetes causing impaired expression of sirtuin-1 [281], a protein that protects mitochondria from damage [282]. In the past years over 47 miRNAs have been found differential between patients with diabetes and patients with DR, five of them, including miR-21 implicated in angiogenic processes, have been considered as biomarkers for early detection of such microvascular complication [283,284].

miRNAs associate with groups of proteins known as Argonaute proteins to silence the target gene. Argonaute proteins are usually found scattered in the cytoplasm, in oxidative stress conditions they tend to gather around stress granules, an action mediated by miRNAs [285]. On the other hand, argonaute2 is enhanced by hypoxia which leads to an inhibited maturation of miRNAs that respond to stress as defense mechanisms [286]. Previously mentioned miRNAs have been found in serum, some of them and others are found in vitreous humor and aqueous humor in PDR as seen in Table 2. miR-126 has recently been considered as a potential serum biomarker for PDR nonetheless, other studies have shown that the concentration of miRNAs in eye matrices with PDR differs even two times fold from those in serum [287].

H-19, a lncRNA that prevents glucose-induced endothelial mesenchymal transition in the retina, is downregulated in hyperglycemic conditions leading to activation of endothelial transition via TFGF-beta [288]. A recent study has found that lncRNA AK077216 is downregulated in subjects with DR independently from glycemic conditions, this lncRNA is able to inhibit ARPE-19 cells apoptosis via inhibition of miR-383 proapototic function [289]. The proliferation promoting lncRNA BANCR has also been found overexpressed in patients with DR which may play a role as a potential therapeutic target [290,291]. According to Awata and cols. there is an association between susceptibility to DR and lncRNA RP1-90L14 [91].

Studies have shown that these alterations occur early in hyperglycemic states in retinal and endothelial cells [292], furthermore; they are maintained even after returning to normoglycemia [293,294,295]. Since non-coding RNA, specially miRNAs have a regulatory function, relatively prompt response to alterations in the body and, more importantly, such responses are maintained for a period of time after glycemic control they represent a potential therapeutic target in DR to delay its progression. Nevertheless more studies are needed so far.

## 7. Conclusions

In the review article, we summarize the studies and take information that contributed to elucidating the most widely used biomarkers that produce more information about the different stages of diabetic retinopathy. Currently, there is still no definitive marker for the detection of early stages of DR or even one that can be detected before retinopathy develops. The reviewed studies have suggested a large number of potential markers, however, it must be taken into account that the multifactorial etiology of DR further complicates the detection strategy, so instead of a single definitive marker, the use of a panel or set of markers representing each affected aspect in DR. For this, it is necessary to choose a biological matrix that meets characteristics such as accessibility, suitability and representativeness of the disease state. At the ocular level, there are very few biomarkers whose measurement has FDA approval. Most of the biomarkers are exploratory in nature; however. they are very useful for solving issues during drug development. Tears, aqueous humor, and vitreous are among the fluid ocular matrices most frequently used for human biomarker evaluation. The use of these matrices has posed difficulties, including non-standardized collection processes and little available sample volume. For this reason, it is of great importance to continue working on studies that provide more information to establish optimized processing and analysis methods in order to make the most of the information obtained from the measurement of biomarkers in them. Large-scale prospective multicenter studies are necessary to be able to accurately determine the veracity and reliability of various biomarkers in the early stages of DR. Therefore, the OS associated with chronic hyperglycemia plays a central role in the stimulation and alteration of the molecular and biochemical signaling pathways along with cellular damage involved in the DR, the evidence suggests that metabolic defects that alter epigenetic substrates they will also affect epigenetic chromatin modifications. The epigenetic alterations that were discussed play a critical role in the pathogenesis of DR, further analysis and comparison of the results that have been obtained so far is necessary to also consolidate the path towards the discovery of new treatments and therapeutic strategies.

## Figures and Tables

**Figure 1 antioxidants-09-00891-f001:**
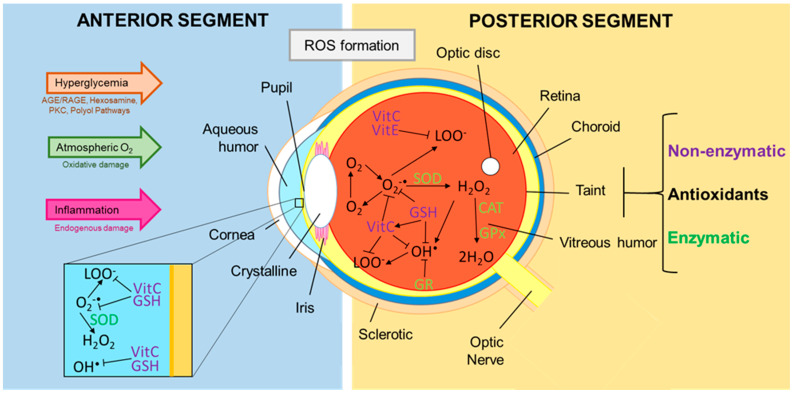
Reactive oxygen species and antioxidants in the eye. In the eye, chronic hyperglycemic state, atmospheric O2, and inflammation processes alter different metabolic pathways which stimulates the formation of reactive oxygen species (ROS) in the anterior and posterior segment, starting with oxygen (O2) to which the addition of one electron forms the superoxide anion radical (O2-•), the dismutation of this molecule by superoxide dismutases (SOD) forms hydrogen peroxide (H2O2), and the breakdown of this molecule can generate hydroxyl radical (OH•) which is highly reactive. In addition the reaction of O2- • or OH• radical with polyunsaturated fatty acids generates the peroxyl radical (LOO•). The formation of this radicals can be countered by enzymatic and non-enzymatic antioxidants like vitamin C, vitamin E, glutathione (GSH), glutathione peroxidase (GPx), superoxide dismutase (SOD), catalase (CAT), and glutathione reductase (GR), among others. (Modified from ref. [24]).

**Figure 2 antioxidants-09-00891-f002:**
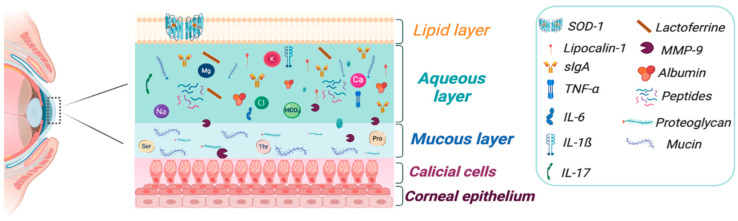
Main components in tear film. The wide variety of components in each layer of the tear film provides function, nutrition, and protection to the ocular surface. However, it is susceptible to change their composition due to oxidative stress and inflammatory processes that involve the eye structures, which makes it an easy access ocular matrix to identify these changes measuring levels of components as biomarkers. SOD1: superoxide dismutase-1, Ser: serine, Thr: Threonine, Pro: Proline, IL: Interleukin, sIgA: surface Immunoglobuline A, MMP-9: matrix metalloproteinase-9, TNF-α: Tumor necrosis factor-α. Image created with BioRender.com.

**Table 1 antioxidants-09-00891-t001:** Biomarker levels in tear, aqueous, and vitreous humor. Levels of the markers of oxidative stress, inflammation, and apoptosis measured in tear, aqueous, and vitreous humor in different ocular conditions, in addition to PDR, differentiating the different methodologies used. PDR: proliferative diabetic retinopathy, DED: dry eye disease, TAC: total antioxidant capacity, LPO: lipoperoxides, MDA: malondialdehyde, GSH: glutathione, GPx: glutathione peroxidase, SOD: superoxide dismutase, 8-OHdG: 8-hidroxi-2-deoxiguanosine, 8-IPGF: 8-isoprostaglandins, NO: nitric oxide, IL-6: interleukin-6, IL-8: interleukin-8, IL-10: interleukin-10, TNF-α: Tumour necrosis factor alpha, VEGF: vascular endothelial growth factor, EGF: epidermal grow factor, MMP-9: metalloproteinase-9, FN-kβ: Nuclear factor kβ, T1DM: type 1 diabetes mellitus, T2DM: type 2 diabetes mellitus.

Profile	Biomarker	Matrix	Pathology	Levels	Method	References
Oxidative Stress	TAC	Aqueous	PDR	0.55 ± 0.28 µmol Trolox/g *	Radical absorbance capacity assay	[104]
		Vitreous	PDR	0.19 ± 0.10 µmol Trolox/g *	Radical absorbance capacity assay	[104]
	LPO	Vitreous	PDR	Male: 145.8 ± 6.3 µM *	Colorimetric assay	[105]
				Female: 135.6 ± 10.9 µM *		
	MDA	Tears	DR	95 µM	Capillary electrophoresis	[106]
		Aqueous humor	Cataract	0.1 ± 0.1 µmol/L *	TBARS method	[107]
		Vitreous	PDR	Male: 101.3 ± 7.6 nmol/mL *	Colorimetric assay	[105]
				Female: 87.6 ± 18.4 nmol/mL *		
	GSH	Tears	DR	107 µM	Chromatography electrochemical	[24]
	GPx	Aqueous	Glaucoma	18.4 ± 2.5 U/mL *	-	[101]
		Aqueous	Cataract	6.1 ± 0.6 U/mL *	-	[101]
	SOD	Tears	DR	1–32 U/mg	Spectrophotometry direct	[45,108]
		Vitreous	PDR	Male: 30.5 ± 2.5 U/mL *	Colorimetric Enzyme assay	[105]
				Female: 28.5 ± 3.8 U/mL *		
	8-OHdG	Aqueous	Cataract	311.6 ± 127.7 µg/mL *	ELISA	[109]
		Aqueous	Myopic	212.5 ± 103.2 µg/mL *	ELISA	[109]
	8-IPGF	Aqueous	Exfoliation syndrome	2429 ± 2940 pg/mL *	Immunoassay	[110]
		Aqueous	Cataract	529.1 ± 226.8 pg/mL *	Immunoassay	[110]
		Aqueous	Diabetic cataract	624 ± 95.7 pg/mL *	ELISA	[111]
	ON	Aqueous humor	PDR	19.43 ± 8.75 µM *	Colorimetric assay	[95]
		Vitreous humor	PDR	0.524 ± 0.27 µM *	spectrophotometric	[112]
			T1DM		Griess reaction	
		Vitreous humor	PDR	0.383 ± 0.17 µM *	spectrophotometric	[112]
			T2DM		Griess reaction	
	L-tyrosine	Tears	DR	45 µM	Chromatography electrochemical	[24]
	L-cysteine	Tears	DR	48 µM	Chromatography electrochemical	[24]
	Ascorbic acid	Tears	DR	665 µM	Chromatography electrochemical	[24]
	Uric acid	Tears	DR	328 µM	Chromatography electrochemical	[24]
Inflammatory	IL-1β	Tears	DR	16.7 ± 3.2 pg/mL *	Multiplex assay Bio-Plex system	[113]
	IL-6	Tears	DR	63.3 ± 12.3 pg/mL *	Multiplex assay	[113]
					Bio-Plex system	
		Tears	DED	26.25 ± 5.20 pg/mL *	Multiplex bead assay	[114]
		Aqueous humor	DR	40.64 ± 16.52 pg/mL *	Multiplex bead immunoassay	[115]
		Aqueous humor	PDR	37.19 pg/mL (3.992–4577.38) **	Immunology Multiplex Assay	[83]
		Vitreous fluid	PDR progression	347.2 pg/mL (26.2–758.6) **	ELISA	[116]
		Vitreous fluid	DR	42.29 ± 10.94 pg/mL *	ELISA	[117]
		Vitreous	DR	64.2 ± 10.4 pg/mL *	Immunoassay	[118]
	IL-8	Aqueous humor	DR	42.20 ± 33.03 pg/mL *	Multiplex bead immunoassay	[115]
		Aqueous humor	PDR	25.28 pg/mL (13.21–184.62) **	Immunology Multiplex Assay	[83]
		Aqueous humor	PDR	76.55 ± 10.88 pg/mL *	ELISA	[119]
		Vitreous humor	PDR	63.55 ± 10.74 pg/mL *	ELISA	[119]
	IL-10	Aqueous humor	DR	0.24 ± 0.16 pg/mL *	Multiplex bead immunoassay	[115]
		Aqueous humor	Fuchs’ uveitis	11.70 ± 6.60 pg/mL *	ELISA	[120]
		Aqueous humor	Behcet’s uveitis	7.23 ± 1.73 pg/mL *	ELISA	[120]
		Vitreous humor	PDR	224.789 ± 43.801 pg/mL *	ELISA	[121]
		Vitreous	DR	4.43 ± 0.4 pg/mL *	Immunoassay	[118]
	IL-17A	Tears	DED	454.67 ± 37.70 pg/mL *	Multiplex bead analysis	[114]
	TNF-α	Tears	NPDR	1.2–5.5 pg/mL	ELISA	[122]
		Tears	PDR	9.2–21.7 pg/mL	ELISA	[122]
		Aqueous humor	DR	4.04 ± 1.83 pg/mL *	Multiplex bead immunoassay	[115]
		Aqueous humor	PDR	84.35 ± 30.82 pg/mL *	CBA technique	[123]
		Vitreous fluid	DR	155.8 ± 82.0 pg/mL *	ELISA	[124]
	IFN-y	Tears	DR	1957.50 ± 166.1 pg/mL *	Multiplex assay Bio-Plex system	[113]
	VEGF-A	Aqueous humor	DR	357.02 ± 84.25 pg/mL *	Multiplex bead immunoassay	[115]
	VEGF	Tears	DR	270.7 ± 40.2 pg/mL *	Multiplex assay Bio-Plex system	[113]
		Aqueous humor	PDR	211.62 pg/mL (48.10–1990.98) **	Immunology Multiplex Assay	[83]
		Vitreous fluid	PDR progression	1789.2 pg/mL (198.5–3436.8) **	ELISA	[116]
		Vitreous	DR	731.20 ± 222.72 pg/mL *	ELISA	[117]
		Vitreous	DR	1491.0 ± 183.1 pg/mL *	ELISA	[125]
	EGF	Tears	DED	1318.9 ± 6 835.0 pg/mL *	Milliplex bead assay	[126]
Apoptosis	MMP-9	Tears	DED	40 ng/mL	Immunoplex	[74,127]
		Aqueous	PDR	160.3 ± 39.5 AU/mL *	Zymographic analysis	[128]
		Vitreous	Macular hole	113.9 ± 229.7 AU/mL *	Zymographic analysis	[129]
	Cytochrome-C	Tears/Cell culture	DED	-	Cell culture	[130]

* Data are expressed as the mean ± SD. ** Median values and 5th and 95th percentile values in pg/mL.

**Table 2 antioxidants-09-00891-t002:** miRNA’s found in human eye fluids of patients. A comparison of some of the most commonly found miRNAs in vitreous or aqueous humor from patients with and without proliferative diabetic retinopathy. PDR: proliferative diabetic retinopathy, NDM: non-diabetic, PVD: proliferative vitreoretinal disease, MH: macular hole without any other condition, DME: diabetic macular edema.

MiRNA	Role	Matrix	Comparison	Result	Author(s)
miR-200b	Angiogenesis promotion [167]	Vitreous humor	PDR vs. NDM	Higher	Gomaa A, 2017 [168]
miR-21	Fibrosis and inflammation promotion [169]	Vitreous humor	PVD vs. MH	Higher	Usui-Ouchi A, 2016 [170]
miR-15a	Angiogenesis inhibition [171]	Vitreous humor	PDR vs. MH	Higher	Hirota K, 2015 [172]
	Pro-inflammatory signaling inhibition [173]	Aqueous humor +	DME +		+ Cho Heeyoon,2020 [174]
miR-320	Apoptosis regulation and angiogenesis repression [175]	Vitreous humor	PVD vs. MH	Higher	Usui-Ouchi A, 2016 [170]
miR-320a miR 320b	Angiogenesis repression [176]	Vitreous humor	PDR vs. MH	Higher	Hirota K, 2015 [172]
miR-184	Apoptosis promotion [177]	Aqueous humor	PDR vs. Cataract	Higher	Chen S, 2019 [178]
miR-93 miR-93-5p	Proliferation and angiogenesis promotion [179]	Vitreous humor	PDR vs. NDM	Higher	Hirota K, 2015 [172]
		Aqueous humor	PDR vs. Cataract		Chen S, 2019 [178]
miR-29a	Angiogenesis inhibition [180]	Vitreous humor	PDR vs. MH	Higher	Hirota, K 2015 [172]
		Aqueous humor +	Cataract +		+ Wecker T, 2016 [93]
miR-16 miR-16-5p	Tumor suppression [181]	Vitreous humor	PVD vs. MH	Higher	Usui-Ouchi A, 2016 [170]
		Aqueous humor +	Cataract +		Wecker T, 2016 [93]
miR-23a	Senescence promotion [182]	Vitreous humor	PDR vs. MH	Higher	Hirota K, 2015 [172]
miR-126	HMGB1 and VCAM-1 regulation [173]	Aqueous humor	PDR vs. Cataract	Lower	Chen S, 2019 [178]
Let-7e	Proliferation inhibition [183]	Vitreous humor	PVD vs. MH	Lower	Usui-Ouchi A, 2016 [170]
miR-204	Apoptosis promotion [184]	Vitreous humor	PVD vs. macular hole	Lower	Usui-Ouchi A, 2016 [170]

+ Biomarker found but not compared to another matrix or group.

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
