# Peer review of "Importance of the Use of Oxidative Stress Biomarkers and Inflammatory Profile in Aqueous and Vitreous Humor in Diabetic Retinopathy"

_antioxidants, 2020, doi:10.3390/antiox9090891_

Round 1

Reviewer 1 Report

Review for ‘Importance of the use of oxidative stress biomarkers 2 and inflammatory profile in aqueous and vitreous 3 humor in Diabetic Retinopathy’. This is a very long ( not brief as mentioned in the conclusion) review looking into possible oxidative stress biomarkers in DR. I would recommend that the authors reformat the paper so that the readers would able to read it more easily. There are multiple paragraphs that are redundant and duplicated as they write the paper according to the anatomy of the eye such as tear/AH/Vitreous then also subdividing it to oxidative stress/inflammation/apoptosis.

First, the tear although could be a biomarker, has so far been not relevant to DR but rather associated with the complications such as neurotrophic ulcer seen in DM patients.

Secondly AH and Vitreous results are very similar so there are a lot of duplicated data which makes it very difficult to read through.

Third, in 2/ Diabetic Retinopathy section should be rewritten correctly

Fourth, 3. Role of oxidants and antioxidants in the pathogenesis of the eye with diabetic retinopathy does not contain roles of antioxidant in DR. if they want to write about the oxidants and antioxidants in the eye as an introduction, I would rename this chapter and do a overview of the above in general eye conditions other than DR and take away all the reviews for dry eye, glaucoma that is dispersed everywhere through out the paper.

Fifth, reorganize the paper such as

  • Method of measuring biomarker in DR
  • Biomarkers in DR
    • Oxidative stress: ROS and relevance with DR
    • Inflammation: cytokines and relevance with DR
    • Apoptosis

  1. Line 83 : NPDR could be symptomatic if associated with macular edema and should be repharased
  2. Line 83-85: NPDR is usually asymptomatic, however, 83 untreated cases tend to progress to PDR, which is generally accompanied by impaired visual 84 acuity[15].
    1. NPDR status it self does not need treatment for the eye, just controlling DM better and monitoring for progression of retinopathy, PDR does not necessary lead to VA loss but is associated with complications such as vitreous hemorrahge or tractional retinal detachment that could lead to vision loss
    2. Inappropriate reference
    3. Please rephrase and find appropriate reference
  3. Line 85: The PDR stage is a consequence of ischemic conditions that arise due to obstruction in the 85 NPDR stage[2].- do not understand what the authors are conveying
  4. Line 91: gradual loss of vision- could be acute or gradual
  5. Line 112-114: The main mechanism of DR pathogenesis is due to alterations in the retinal microvasculature as 112 a result of hyperglycemia. The high concentration of glucose in the blood for a long time generates 113 numerous molecular changes[22].: not needed
  6. Line 111. 3.1 Oxidative stress and the damage it causes to the eye: mainly explains oxidative stress and high level of H2 O 2 in cataract eyes. Please add more details of level of oxidative stress in the retina, especially associated with DR
  7. Line 317: Diabetic 316 Retinopathy Early Treatment Study (ETDRS) score including genotypes or tumor markers[78].: please rephrase: ETDRS scores does not include genotype or tumor markers
  8. Line 335-337: already explained in prior paragraphs. Not necessaryneeded here
  9. Line 345-356: performed in the clinic to facilitate 345 sampling [76]: it is an invasive procedure ( not small) and can contain significant eye complications.
  10. 1.1 tears, oxidative stress: please limit to explaining those associated with diabetic retinopathy not with dry eye or other condition
  11. Line 434,444: would discuss more about the studies regarding DR. also rephrase ‘studies has been 434 reported a positive correlation between the AGE products with tear lactoferrin, this suggests the 435 prediction on the progression of retinal damage in patients with DR[92, 93]. ‘ given controversial results between 2 studies, it does not give us any suggestion that the progression of DR can be predicted with AGE products in tear lactoferrin
  12. Line 440-450: This plays a role in the induction of neovascularization with stimulation of 449 pro-angiogenic factor and modulation of cytoskeleton, this was examined in vitreous and plasma of 450 patients with DR, with levels significatively increased in PDR. : this could go to the vitreous section
  13. 1.2 inflammation: I would start off with that DM patients with advanced diabetic retinopathy have more dry eye issues which could cause severe vision loss due to cornea ulcer or neurotrophic ulcer. There are many studies looking into neurotrophic ulcers in diabetes patients.
  14. Line 488-491: Nevertheless, not only in diabetes mellitus the analysis of proinflammatory cytokines can be 488 useful, also in other frequent alterations of the ocular surface such as glaucoma, dry eye disease, 489 allergic conjunctivitis, keratoconjunctivitis, rosacea and ophthalmopathy in hyperthyroidism also 490 have been studied as biomarkers, as well in the post-surgical evaluation of refractive surgery, 491 patients who wear contact lenses and infections [84].: no need for this sentence, please focus on DM and biomarkers
  15. Line 512-524 The tear film is an accessible and minimally invasive matrix that needs to be more study and 521 used as source for a wide variety of diseases that could be easily used in clinical practice and the 522 metabolomics resources that are developing might facilitate the standardization on levels of a wide 523 variety of cytokines, enzymes and metabolic residues that can be useful as therapeutics targets that 524 can improve the diagnosis and treatments for many patients with ocular and systemic diseases.: redundant
  16. 1.3.Apoptosis : any studies specifically associated with DR?
  17. Line 527: delete Therefore,
  18. 21 oxidative stress: OS and antioxidant status have been implicated in multiple eye diseases such as glaucoma and 553 DR. Once OS occurs, the levels of ROS exceed antioxidant defenses[121]. Multiple OS markers can be 554 detected in AH have been reported in the literature.- please restrict to findings associated with diabetes retinopathy this paragraph also describes multiple findings from oxidative stress associated with glaucoma, which is not the main subject of this review.
  19. 2.2 Inflammation: exhaustively long laying out findings from study. Please rewrite so that is is easier for readers to understand. Biomarkers such as cytokines related to inflammation in DR with AH or vitreous  are the most researched area and hence most significant findings that we find with DR.
  20. 2.3 Apoptosis Line 685-688 redundant : please summarize what biomarkers in AH can we get that represents apoptosis.
  21. 3 vitreous humor: again here to there are many redundant exhaustive information regarding biomarkers representing oxidative stress, inflammation and apoptosis which is very similar to the results with the AH; again would recommend rewriting/ reformatting the paper for the readers to easily understand the biomarkers that have been studied in the past

Reviewer 2 Report

General comments:

            Diabetic retinopathy (DR) is a major microvascular complication of diabetic patients and the key factor to cause blindness worldwide.  This manuscript’s primary purpose is to review the role of various biomarkers related to oxidative stress, inflammation, apoptosis, and metabolic memory of oxidative stress from different biological matrices obtained from the eye in DR. Although this manuscript is well-written, there are some significant issues needed to be addressed.

Comments to the authors:

Major comments:

  1. In Abstract, the authors only generally discuss the topic of oxidative stress biomarkers and inflammatory profile of DR. It is vital for the authors to highlight some of the key biomarkers related to DR, which will give readers a good impression about the significance of this manuscript.

  1. On lines 76-78, the authors used “it is estimated” repeatedly, and it is needed to change the sentences.

  1. On lines 112-114, the authors need to delete “The high concentration of glucose in the blood for a long time” because they already mentioned hyperglycemia in the previous sentence.

  1. On line 186, the authors discuss COX pathway, but they only mentioned PGE2 and 8-iso-PGF2α. It is well known that the COX pathway contains PGI2, PGE2, PGF2a, and TXA2, and the authors need to consider the role of these lipid metabolites in oxidative stress of the retina.

  1. On line 305, the authors need to delete the sentence “A biomarker can be defined as a characteristic that is objectively---” because they already define biomarker several times in the previous sections.

  1. On line 635, the authors stated “the NO levels found in the aqueous humor were 635 significantly higher in patients with PDR”. Since NO is unstable in vivo, the authors need to indicate what kind of parameter to indicate NO levels.

  1. On line 685, the authors need to delete “Apoptosis is a process by which cells regulate self-destruction in multicellular organisms” because apoptosis has been discussed in previous sections.

  1. On line 721, the authors indicated that nuclear actor-kβ (NF-kβ) is a vital nuclear factor in apoptosis and cellular neovascularization. Still, they did not discuss which factors activate NF-kβ.  It is essential for them to review factors, such as eicosanoids, activate NF-kβ. 

Round 2

Reviewer 1 Report

The authors revised the manuscript as recommended. 

Reviewer 2 Report

No further comments.